# Look Less, Reason More: Rollout-Guided Adaptive Pixel-Space Reasoning

## Abstract

Vision-Language Models (VLMs) excel at many multimodal tasks, yet they frequently struggle with tasks requiring precise understanding and handling of fine-grained visual elements. This is mainly due to information loss during image encoding or insufficient attention to critical regions. Recent work has shown promise by incorporating pixel-level visual information into the reasoning process, enabling VLMs to access high-resolution visual details during their thought process. However, this pixel-level information is often overused, leading to inefficiency and distraction from irrelevant visual details. To address these challenges, we propose the first framework for adaptive pixel reasoning that dynamically determines necessary pixel-level operations based on the input query. Specifically, we first apply operation-aware supervised fine-tuning to establish baseline competence in textual reasoning and visual operations, then design a novel rollout-guided reinforcement learning framework relying on feedback of the model's own responses, which enables the VLM to determine when pixel operations should be invoked based on query difficulty. Experiments on extensive multimodal reasoning benchmarks show that our model achieves superior performance while significantly reducing unnecessary visual operations. Impressively, our model achieves 73.4% accuracy on HR-Bench 4K while maintaining a tool usage ratio of only 20.1%, improving accuracy and simultaneously reducing tool usage by 66.5% compared to the previous methods.

## 1 Introduction

Vision-Language Models (VLMs) have achieved remarkable progress, leveraging large language models and powerful vision encoders. Modern VLMs, such as GPT-4 (Hurst et al., 2024), Qwen-VL (Bai et al., 2025; Wang et al., 2024a), InternVL (Zhu et al., 2025; Wang et al., 2025b) and LLaVA (Li et al., 2024; Liu et al., 2023; 2024), can perform sophisticated visual understanding and reasoning tasks (Shen et al., 2025). However, VLMs frequently encounter difficulties in capturing fine-grained visual elements, largely because of information loss in the image encoding process or the limited allocation of attention to critical regions (Ge et al., 2024; He et al., 2024). Recently, advanced models (Su et al., 2025a; Wang et al., 2025c; Zhang et al., 2025b; Zheng et al., 2025; Zhou et al., 2025) have been proposed, which are capable of executing pixel-level operations—an ability we refer to as **pixel-space reasoning**. By zooming into specific image regions, these models can selectively focus on critical areas when the original image is too complex.

Existing models or frameworks that allow pixel-level operations can be broadly categorized into pipelining and end-to-end strategies. Pipelining approaches (Hu et al., 2024b;c; Lu et al., 2025; Liu et al., 2025; Li et al., 2025e) typically consist of multiple components, such as a predefined cropping tool or auxiliary feature extractors. While computationally efficient, they tend to leverage visual information more passively, rather than being actively shaped by the model's reasoning needs. Therefore, they often fail to capture subtle but essential visual cues, especially in tasks requiring spatial reasoning or fine-grained perception. End-to-end strategies, in contrast, enable the model to actively manipulate visual inputs through pixel-level operations (Zheng et al., 2025; Zhou et al., 2025; Su et al., 2025b), such as zooming into specific regions.

Despite the flexibility of existing end-to-end methods (Wang et al., 2025a; Zhang et al., 2025a; Huang et al., 2025b), they often encourage the application of pixel-level operations regardless of

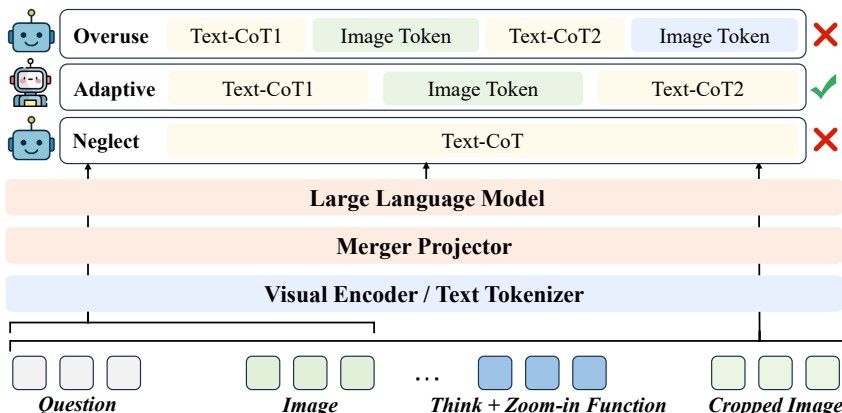

Figure 1: Comparison of different reasoning strategies. The "Overuse" strategy unnecessarily incorporates pixel-level operations, leading to inefficiency and potential distraction. The "Neglect" strategy relies solely on pure textual CoT reasoning, failing to engage with critical fine-grained visual details. Our "Adaptive" strategy achieves a balance by intelligently deciding whether to perform pixel-level operations based on the specific query, optimizing both accuracy and efficiency.

whether the operations are actually needed. This overuse of pixel-level operations causes the following weaknesses: 1) **Computational inefficiency**: the frequent encoding of parts of the images requires additional time and slows down the inference speed. 2) **Learning difficulties**: cropped images occupy substantial context space, potentially introducing noise and causing error propagation in the sequential generation process, particularly when cropped regions are irrelevant to the query. Ideally, a model should adaptively decide when to invoke pixel-level operations to focus more on relevant regions, and when a pure textual chain of thought (CoT) (Wei et al., 2022) alone suffices, thereby striking a balance between accuracy and efficiency. One straightforward solution is to have human experts manually label whether each query requires pixel-level operations, thereby providing additional supervision to guide the VLMs. However, this approach is both tedious and costly, making it impractical at scale. This naturally raises the question: can VLMs learn to apply pixel-level operations only when necessary, without relying on additional, predefined labels?

To address this, we propose the first framework for adaptive pixel-space reasoning that equips VLMs with the ability to dynamically determine the necessity of pixel-level operations. Since current open-source VLMs are rarely trained with pixel-level operations, we begin with **operation-aware supervised fine-tuning (SFT)** (§4.1), which provides the model with baseline competence in answering visual-related questions with or without pixel-level operations following specifications in the query. Afterwards, we design a novel **rollout-guided reinforcement learning (RGRL)** framework (§4.2) to enhance adaptive pixel-space reasoning capability. Unlike the conventional RL approach, which typically only promotes accuracy and encourages the frequency of tool usage, we carefully design the reward assignment strategy to encourage the VLMs to leverage pixel reasoning only when it is beneficial. Our rollout-guided RL framework consists of two complementary components: (1) Pixel Necessity Rollouts, VLMs are explicitly required to produce answers both with and without pixel operations. The relative success rates provide implicit pixel operation necessity indicating whether pixel-level operations are beneficial for the query. (2) Adaptive Rollouts, which encourage VLMs to autonomously decide whether and how to apply pixel operations. Rewards are determined not only by the correctness of the responses, but also by their consistency with the necessity estimated in the previous rollouts. In this way, we promote efficient and robust adaptive pixel-space reasoning leveraging only the VLM's own responses.

Extensive experiments show that our framework outperforms both general-purpose VLMs and strong tool-augmented baselines, achieving the highest average accuracy while minimizing unnecessary visual operations (§5.2). Specifically, our framework achieves 73.4% accuracy on HR-Bench 4K (Wang et al., 2024b) while maintaining a tool usage ratio of only 20.1%, improving accuracy and simultaneously reducing tool usage by 66.5% compared to the previous methods. Qualitative analysis further validates that our model can adaptively identify relevant visual regions and perform pixel operations only when contextually appropriate (§5.4).

In summary, this work makes three key contributions: **1)** we introduce the first framework that enables adaptive pixel-space reasoning, allowing VLMs to determine when pixel-level operations

are necessary rather than applying them indiscriminately; **2)** our training framework does not rely on any external pixel-level supervision or hand-crafted rules, allowing the model to estimate the necessity of pixel-level operations directly from its own reasoning process; **3)** we achieve superior performance compared to existing baselines across five multimodal reasoning benchmarks while simultaneously improving reasoning accuracy and tool efficiency.

## 2 RELATED WORK

**Vision-Language Models.** Vision-Language Models (VLMs) have evolved from early pipelines connecting visual encoders to frozen language models into more unified architectures trained with joint objectives. Representative frameworks such as BLIP-2 (Li et al., 2023) and LLaVA (Liu et al., 2023) employ connector modules—either projection layers (Li et al., 2025a; Cha et al., 2024) or attention-based adapters (Hu et al., 2023; Song et al., 2024)—to align image features with text embeddings, enabling tasks such as visual question answering and instruction following (Li et al., 2025c;d). Later research addresses perception bottlenecks, enhancing encoder capacity (Shen et al., 2024) or introducing dynamic resolution strategies (Anghelone et al., 2023). Open-source series (Wang et al., 2024a; Bai et al., 2025) and large-scale systems like Flamingo (Alayrac et al., 2022) and mPLUG-Owl (Ye et al., 2023; 2024) demonstrate competitive performance across multimodal benchmarks. Despite these developments, most models remain perception-centric, leaving room for improvements in complex multimodal reasoning.

**Textual-space VLM Reasoning.** Textual-space reasoning refers to approaches where VLMs improve reasoning by producing pure textual CoT, without directly manipulating pixels. Early works (Chen et al., 2023; Zhang et al., 2023) showed that inserting CoT steps enhances visual question answering. Follow-up methods refined this paradigm by improving rationale quality through self-consistency (Tan et al., 2023), dynamic routing (Aytes et al., 2025; Hu et al., 2025), or multi-image (Zhang et al., 2024; Xie et al., 2025) and relation-aware reasoning. Other directions emphasized interpretability via staged reasoning (Zheng et al., 2023) or automatic rationale generation to reduce annotation cost (Ma et al., 2024; Luo et al., 2024). Despite these advances, textual-space reasoning relies on static image embeddings and lacks mechanisms to adaptively refine visual evidence, which motivates pixel-space reasoning approaches.

**Pixel-space VLM Reasoning.** Pixel-space reasoning, or "thinking with images," refers to approaches where models actively manipulate visual inputs—such as cropping, masking, or sketching—rather than relying solely on pure textual CoT. Early attempts (Liu et al., 2025; Huang et al., 2025a; Lu et al., 2025) followed predefined workflows or required auxiliary annotations like spatial layouts, attributes, or external knowledge, which limited their generality. More recent tool-augmented frameworks (Su et al., 2025a; Wang et al., 2025c; Zhang et al., 2025b; Zheng et al., 2025; Zhou et al., 2025) take a step toward interactive multimodal reasoning by enabling direct pixel-level operations. However, they often lack principled strategies for deciding when and how to invoke these operations (Feng et al., 2025; Li et al., 2025b), leading to inefficiency or distraction. These limitations motivate adaptive mechanisms that dynamically balance accuracy and efficiency. Unlike prior work that either hard-codes tool usage or overlooks its cost, our method explicitly learns when pixel-level operations are beneficial, achieving adaptive visual reasoning.

## 3 PROBLEM FORMULATION

Multimodal reasoning involves solving queries that require varying degrees of pixel-level operations. While some queries can be accurately addressed using the model's pure textual CoT, others demand focused pixel-level exploration to extract fine-grained information. This motivates *adaptive pixel-space reasoning*, where the model dynamically determines whether to invoke a pixel-level operation.

Formally, let $\mathbf{x} = [V, L]$ denote a vision-language query, with $V$ representing the visual input and $L$ the textual instruction. The model generates a reasoning trajectory $\mathbf{y} = [y_1, \ldots, y_n, \hat{a}]$, where each step $y_t$ can be either a pure textual CoT or a zoom-in operation, and $\hat{a}$ is the model's final predicted answer. The zoom-in operation extracts high-resolution information from a specified region of $V$, which is then incorporated into the subsequent reasoning steps: $y_t \leftarrow \text{concat}(y_t, f_{\text{zoom-in}}(y_t))$, where $f_{\text{zoom-in}}(y_t)$ denotes the high-resolution visual features acquired by the zoom-in operation.

To evaluate solution correctness, we compare the predicted answer $\hat{a}$ with the ground-truth answer $a^*$ and define the reward:

$$r_{\text{correct}}(\mathbf{x}, \mathbf{y}) = \begin{cases} 1 & \text{if } \hat{a} = a^*, \\ 0 & \text{otherwise.} \end{cases} \tag{1}$$

The overall objective of RL training can then be written as

$$\max_{\theta} \mathbb{E}_{\mathbf{x}\sim\mathcal{D},\, \mathbf{y}\sim\pi_\theta(\mathbf{y}|\mathbf{x})}\Big[ R(\mathbf{x}, \mathbf{y}) \Big], \quad R(\mathbf{x}, \mathbf{y}) = r_{\text{correct}}(\mathbf{x}, \mathbf{y}) + \lambda\, r_{\text{pixel}}(\mathbf{x}, \mathbf{y}), \tag{2}$$

where $r_{\text{pixel}}(\mathbf{x}, \mathbf{y})$ provides a positive reward if a pixel-level operation improves the final answer $\hat{a}$ and a negative reward if it is unnecessary or detrimental, and $\lambda$ controls the trade-off between correctness and efficiency.

Under this formulation, the model must develop a query-specific adaptive strategy: it should invoke zoom-in selectively, only when pixel-level operations contribute to the final solution. By explicitly considering the benefit of visual operations, the framework encourages accurate, efficient, and robust multimodal reasoning across diverse query complexities.

## 4 METHOD

Existing RL methods for pixel-space reasoning often fail to learn an adaptive strategy, leading to two common failure modes: either an over-reliance on zoom-in or a complete avoidance of it. To address this, we propose an **adaptive rollout-guided RL training framework** that enables dynamic decision-making for visual exploration. Our method consists of two primary stages: operation-aware SFT phase (§4.1) and rollout-guided reinforcement learning (RGRL) phase (§4.2).

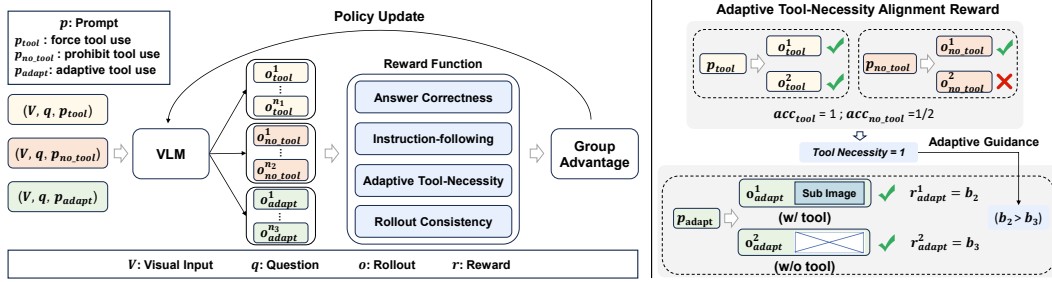

Figure 2: Overview of rollout-guided reinforcement learning. The framework generates rollouts under three prompting modes: forced tool use, prohibited tool use, and adaptive tool use, and these rollouts are rewarded by multiple reward functions. The adaptive tool-necessity alignment reward leverages comparisons between tool and no-tool rollouts to determine pixel tool necessity and guide the adaptive rollout, where the reward is determined by the model's own adaptive reasoning and match of tool necessity. All rewards are aggregated to compute group advantage, which updates the policy to achieve efficient and adaptive visual reasoning.

### 4.1 OPERATION-AWARE SUPERVISED FINE-TUNING

We begin with a supervised training stage on $\mathcal{D}_{\text{SFT}}$, a dataset that not only provides question-answer pairs but also their detailed reasoning trajectories. These trajectories are operation-aware: a portion of them involves explicit pixel-level operations, while others rely purely on textual CoT. By exposing the model to both categories, this stage enables it to establish foundational competence in both pure textual CoT and the proper execution of visual operations. It effectively prepares the model for the more complex adaptive RGRL stage by having it minimize a standard cross-entropy loss:

$$\mathcal{L}_{\text{SFT}} = - \sum_{(\mathbf{x}_i, \mathbf{y}_i) \in \mathcal{D}_{\text{SFT}}} \log P_\theta(\mathbf{y}_i \mid \mathbf{x}_i), \tag{3}$$

where $\mathbf{x}_i$ denotes the input query, $\mathbf{y}_i$ is the reasoning trajectory, and $\theta$ is the model parameters.

## 4.2 ROLLOUT-GUIDED REINFORCEMENT LEARNING (RGRL)

After the SFT training, we transition to rollout-guided RL, where the model learns to achieve adaptive pixel-space reasoning. For each vision-language query $\mathbf{x} = [V, L]$, we perform a total of $N$ reasoning rollouts. These rollouts are strategically divided into two groups: *pixel necessity rollouts*, which evaluate the necessity of zoom-in and provide an implicit tool necessity signal, and *adaptive rollouts*, where the model learns to make its own informed decisions. To control the model's behavior during each rollout, we prepend a specific system prompt in the textual instruction $L$.

### 4.2.1 PIXEL NECESSITY ROLLOUTS

The first $N_{\text{necessity}} = n_1 + n_2$ rollouts are controlled to estimate the query-specific necessity of invoking zoom-in. We achieve this by using different system prompts. For the first $n_1$ rollouts, we use system prompt $p_{\text{tool}}$ to force a tool-use action. For the next $n_2$ rollouts, we use system prompt $p_{\text{no\_tool}}$ to prohibit tool use. This setup provides two distinct performance baselines: one with pixel operation and one with only pure textual CoT. We then compare the average accuracy of these two groups, $acc^{\text{tool}}$ and $acc^{\text{no\_tool}}$, to determine a query-specific adaptive tool necessity. Let $\mathbf{1}_{\text{tool\_necessity}}$ denotes the indicator of the necessity to use pixel-space operations (1 if necessary, 0 otherwise). This tool necessity provides a crucial guidance signal for subsequent learning:

$$\mathbf{1}_{\text{tool\_necessity}} = \begin{cases} 1 & \text{if } acc^{\text{no\_tool}} < acc^{\text{tool}}, \\ 0 & \text{otherwise.} \end{cases} \tag{4}$$

**Instruction-following Reward.** During the pixel necessity estimation phase, we apply an *instruction-following reward* to ensure the model follows the enforced system prompt for the entire reasoning trajectory. Let $\mathbf{z} = [z_1, \ldots, z_m]$ denote the sequence of pixel-level actions in the trajectory, and let $\mathcal{Z}^{\text{prompt}} \subset \{0, 1\}$ be the set of allowed actions according to the current prompt ($\{1\}$ for forced zoom-in, $\{0\}$ for prohibited zoom-in). We define the reward as

$$r_{\text{instr}} = \begin{cases} +b_1, & \text{if } \exists t \text{ s.t. } z_t \in \mathcal{Z}^{\text{prompt}}, \\ -c_1, & \text{otherwise,} \end{cases} \tag{5}$$

where $b_1, c_1 > 0$ are positive constants. That is, the trajectory receives a positive reward if it contains at least one action allowed by the prompt, and a negative reward otherwise.

### 4.2.2 ADAPTIVE ROLLOUTS

The remaining $N_{\text{adaptive}} = n_3$ rollouts allow the model to learn its adaptive strategy. For these attempts, a neutral system prompt $p_{\text{adapt}}$ is used, letting the model freely decide whether to invoke a zoom-in operation. Each adaptive rollout guides the model to learn an efficient, query-specific strategy. For detailed prompts, please refer to Appendix B.

**Adaptive Tool-Necessity Alignment Reward.** This reward encourages the model to align its zoom-in decisions with the query-specific tool necessity obtained from the pixel necessity rollouts. Let $\mathbf{1}_{\text{zoom}} \in \{0, 1\}$ denote whether a zoom-in operation is performed during the thought process (1 if performed, else 0), $\mathbf{1}_{\text{correct}} \in \{0, 1\}$ indicate whether the final answer is correct, and $m = \mathbf{1}[(\mathbf{1}_{\text{zoom}} = 1 \land \mathbf{1}_{\text{tool\_necessity}} = 1) \lor (\mathbf{1}_{\text{zoom}} = 0 \land \mathbf{1}_{\text{tool\_necessity}} = 0)]$ represent whether the zoom decision matches the query-specific necessity ($m = 1$ if matched, else $m = 0$). We define the adaptive tool-necessity alignment reward as:

$$r = \begin{cases} +b_2, & \text{if } \mathbf{1}_{\text{correct}} = 1 \text{ and } m = 1, \\ +b_3, & \text{if } \mathbf{1}_{\text{correct}} = 1 \text{ and } m = 0, \\ -c_2, & \text{if } \mathbf{1}_{\text{correct}} = 0 \text{ and } m = 1, \\ -c_3, & \text{if } \mathbf{1}_{\text{correct}} = 0 \text{ and } m = 0, \end{cases} \tag{6}$$

where $b_2, c_2, b_3, c_3 > 0$ are positive real numbers, with $b_2 > b_3$ and $c_3 > c_2$. Intuitively, the reward separates two factors: (i) whether the zoom decision matches the query-specific necessity, and (ii) whether the final answer is correct. If the model follows the query-specific necessity *and* produces a correct answer, it receives $+b_2$; if it follows the guidance but the answer is incorrect, it

Table 1: Performance and tool usage ratio of models on five multimodal reasoning benchmarks. Numbers in the top row indicate Accuracy (or ANLS for InfoVQA), while the gray numbers in parentheses indicate the corresponding **tool usage ratio** (%). * denotes results reproduced by ourselves, † denotes methods using GPT-4V.

| Model | Size | V* Bench | MMStar | HR-Bench 4K | HR-Bench 8K | InfoVQA | Avg |
|---|---|---|---|---|---|---|---|
| | | | | **Model w/o Tools** | | | |
| GPT-4o | - | 62.8 | 61.6 | 59.0 | 55.5 | 80.7 | 63.9 |
| Gemini-2.0-Flash | - | 73.2 | - | - | - | 86.5 | - |
| Gemini-2.5-Pro | - | 79.2 | - | - | - | 84.0 | - |
| LLaVA-OneVision | 7B | 75.4 | 61.7 | 63.0 | 59.8 | 68.8 | 65.7 |
| DeepSeek-VL | 7B | - | 40.5 | 35.5 | 33.4 | - | - |
| IXC2-4KHD | 7B | - | - | 57.8 | 51.3 | 68.6 | - |
| Video-R1 | 7B | 51.2 | - | - | - | 67.9 | - |
| LongLLava | 13B | 68.5 | - | - | - | 65.4 | - |
| Gemma3 | 27B | 62.3 | - | - | - | 59.4 | - |
| Qwen2.5-VL* | 7B | 73.3 | 63.6 | 67.3 | 64.1 | 78.5 | 69.4 |
| | | | | **Model w/ Tools** | | | |
| IVM-Enhance† | - | 81.2 | - | - | - | - | - |
| SEAL | 7B | 74.8 | - | - | - | - | - |
| PaLI-X-VPD | 55B | 76.6 | - | - | - | - | - |
| Pixel Reasoner* | 7B | 84.3 (80.7) | 63.4 (47.1) | 72.6 (86.6) | 66.1 (87.4) | 83.9 (25.1) | 74.1 (65.4) |
| **Ours** | **7B** | **85.9** $(59.1)_{-21.6}$ | **64.3** $(37.9)_{-9.2}$ | **73.4** $(20.1)_{-66.5}$ | **66.6** $(48.5)_{-38.9}$ | **84.4** $(14.6)_{-10.5}$ | **74.9** $(36.0)_{-29.4}$ |

receives $-c_2$; if it does not follow the guidance but still answers correctly, it receives $+b_3$; otherwise it receives $-c_3$. We evaluate two dimensions—adherence and correctness. Since the reward for being both adherent and correct should exceed that for being correct despite non-adherence, we set $b_2 > b_3 > 0$. Moreover, the case associated with $c_3$ corresponds to simultaneous non-adherence and incorrectness; hence it incurs the largest penalty, with $c_3 > c_2 > 0$. Together, these constraints encourage both correctness and adherence to the tool-necessity guidance.

**Rollout Consistency Reward.** To encourage stable decisions across rollouts of the same query, we penalize inconsistent tool usage among the $N_{\text{adaptive}}$ adaptive rollouts:

$$r_{\text{cons}} = -\gamma \operatorname{Var}(\mathbf{1}_{\text{zoom}}), \quad \gamma > 0. \tag{7}$$

The $\operatorname{Var}(\mathbf{1}_{\text{zoom}})$ measures the variability of tool usage, with lower variance corresponding to more consistent decisions.

### 4.2.3 OVERALL ROLLOUT-GUIDED REWARD

The overall objective is to maximize a unified reward $R$, which is realized differently in the two rollout phases: $R_{\text{necessity}}$ for pixel necessity rollouts and $R_{\text{adapt}}$ for adaptive rollouts.

For the *pixel necessity rollouts*, the reward combines correctness and instruction-following:

$$R_{\text{necessity}} = r_{\text{correct}} + \lambda_{\text{instr}} \, r_{\text{instr}}, \tag{8}$$

where $\lambda_{\text{instr}} > 0$ controls the relative importance of following the prompt versus answering correctly.

For the *adaptive rollouts*, the reward incorporates three components, guiding the model towards an optimal, stable, and adaptive strategy:

$$R_{\text{adapt}} = r_{\text{correct}} + \lambda_{\text{align}} \, r_{\text{align}} + r_{\text{cons}}, \tag{9}$$

where $\lambda_{\text{align}} > 0$ balances the influence of the adaptive tool-necessity alignment reward relative to correctness and consistency.

## 5 EXPERIMENTS

### 5.1 SETUPS

**Training.** We follow Pixel-Reasoner (Su et al., 2025a) and use its datasets, comprising 4k samples for SFT and 7k samples for RL. The base model is Qwen2.5-VL-7B-Instruct (Bai et al., 2025). We adopt Open-R1 (Hugging Face, 2025) for SFT and OpenRLHF (Hu et al., 2024a) for RL. For SFT, we use a batch size of 128 and a learning rate of $1 \times 10^{-6}$, with 10% warm-up steps. For RL, we employ a cosine learning rate schedule with a learning rate of $1 \times 10^{-6}$. Each batch samples 256

prompts, with $N = 16$ rollouts per prompt ($n_1 = 4$, $n_2 = 4$ and $n_3 = 8$), allowing at most 6 pixel-level operations. We provide detailed hyperparameters in the Appendix C.

**Baseline.** We compare our approach with general-purpose and tool-augmented VLMs. The first group includes representative VLMs such as GPT-4o (Hurst et al., 2024), Gemini-2.5 series (Team et al., 2024; Comanici et al., 2025; Team et al., 2023), LLaVA-OneVision (Li et al., 2024), DeepSeek-VL (Lu et al., 2024), InternLM-XComposer2-4KHD (IXC2-4KHD) (Dong et al., 2024), Qwen2.5-VL (Bai et al., 2025), Video-R1 (Feng et al., 2025), LongLLaVA (Wang et al., 2024c), and Gemma3 (Team et al., 2025). These models directly perform reasoning without external tool invocation. The second group consists of tool-augmented models, including Instruction-Guided Masking (IVM-Enhance) (Zheng et al., 2024), Visual-Program-Distillation (PaLI-X-VPD) (Hu et al., 2024c), SEAL (Wu & Xie, 2024), and Pixel Reasoner (Su et al., 2025a), which represents a strong baseline for pixel-space reasoning with its innovative approach to zoom-in visual operations.

**Benchmark.** We evaluate our method across five diverse multimodal benchmarks, covering both fine-grained perception and complex high-level reasoning including V* (V-Star) Bench (Wu & Xie, 2024), MMStar (Chen et al., 2024), HR-Bench (4K/8K) (Wang et al., 2024b) and InfographicVQA (InfoVQA) (Mathew et al., 2022). Among these benchmarks, all adopt Accuracy metrics for evaluation except InfoVQA, which uses the Average Normalized Levenshtein Similarity (ANLS) metric.

## 5.2 Main Results

**Our method achieves consistent superior performance on multimodal reasoning benchmarks, outperforming both general-purpose VLMs and strong tool-augmented systems.** As shown in Table 1, compared to existing baselines, our method achieves the highest average score. Both our method and Pixel Reasoner are trained based on Qwen2.5-VL under comparable data settings. While Pixel Reasoner exhibits performance degradation on MMStar due to indiscriminate pixel-level operations, our method maintains consistent superior performance across all five benchmarks. This demonstrates that our adaptive framework can effectively determine when pixel-level operations are truly necessary, avoiding redundant computations while preserving accuracy.

**Adaptive tool usage significantly reduces unnecessary visual operations without sacrificing accuracy.** We further analyze the tool usage ratio across benchmarks in Table 1. The result shows that our model adaptively balances pure textual CoT and pixel-level operations assistance, achieving an average tool ratio of 36.0%, substantially lower than the existing strong tool-augmented baseline Pixel Reasoner (65.4%). The lower overall average ratio primarily reflects our ability to avoid redundant tool invocations, indicating that the model not only achieves better accuracy but also reduces unnecessary computational overhead during the thought process.

**Adaptive reasoning capabilities emerge through Rollout-Guided RL training.** The task-dependent distribution of the tool usage ratio provides strong evidence that our framework has successfully trained the model to possess adaptive reasoning capabilities. Our model naturally invokes fewer tools on relatively simple benchmarks (e.g., InfoVQA, tool ratio 14.6%) while increasing tool reliance on more challenging benchmarks (e.g., HR-Bench 8K, tool ratio 48.5%), demonstrating that the learned adaptive behavior aligns with the actual reasoning demands of queries. Besides, as shown in Table 2, our RGRL training can effectively correct redundant tool usage patterns learned during SFT. For instance, on InfoVQA, the model initially exhibits excessive tool usage (20.1%) after SFT, but our RL training successfully reduces this to 14.6%, while simultaneously improving accuracy from 73.9% to 84.4%.

## 5.3 Ablation Study

### 5.3.1 Effectiveness of Rollout-Guided RL (RGRL)

We also evaluate our model without the RGRL phase, relying solely on operation-aware SFT. As shown in Table 2, without RL training, the variant exhibits a significantly lower accuracy and higher tool usage compared to our full approach. These results suggest that, while SFT alone provides foundational capability, it lacks the ability to dynamically adjust tool usage based on the complexity of the task. This reinforces the effectiveness of combining operation-aware SFT with RGRL to enhance the model's adaptive decision-making in multimodal reasoning tasks.

Table 2: Ablation study on the effectiveness of rollout-guided RL.

| Model | V* Bench | MMStar | HR-Bench 4K | HR-Bench 8K | InfoVQA | Avg |
|---|---|---|---|---|---|---|
| Ours w/o RGRL | 78.5 (9.4) | 58.4 (39.3) | 66.6 (68.8) | 57.0 (65.3) | 73.9 (20.1) | 66.9 (40.6) |
| **Ours** | **85.9** (59.1) | **64.3** (37.9) | **73.4** (20.1) | **66.6** (48.5) | **84.4** (14.6) | **74.9** (36.0) |

### 5.3.2 COMPARISON OF DIFFERENT TOOL USAGE STRATEGIES

The results of the ablation study, which evaluates our trained model under different tool usage prompts, are shown in Table 3. The first two rows correspond to extreme cases: **All No-Tool**, where the model relies solely on pure textual CoT, and **All Tool**, where the model always uses pixel-level operations. The All No-Tool strategy achieves an average accuracy of 72.4 across the five benchmarks, while the All Tool strategy achieves 72.5. Both are lower than our adaptive method, which reaches an average accuracy of 74.9. The All-Tool strategy underperforms particularly on high-resolution benchmarks such as HR-Bench 8K, showing that excessive reliance on pixel-level operations can be counterproductive. Frequent zoom-in operations lead to redundant cropping, which introduces noisy visual paths and distracts the reasoning process. Similarly, the All No-Tool strategy cannot fully exploit the benefits of visual operations in complex scenarios, as it can't zoom into critical regions and extract fine-grained visual cues. In contrast, our adaptive method determines dynamically when tool usage is beneficial, leading to the highest accuracy on all five benchmarks.

Table 3: Ablation study of different tool usage strategies.

| Model | V* Bench | MMStar | HR-Bench 4K | HR-Bench 8K | InfoVQA | Avg |
|---|---|---|---|---|---|---|
| All No-Tool | 81.2 | 63.8 | 70.9 | 63.8 | 82.1 | 72.4 |
| All Tool | 83.2 | 63.5 | 71.3 | 62.6 | 81.7 | 72.5 |
| **Ours** | **85.9** | **64.3** | **73.4** | **66.6** | **84.4** | **74.9** |

### 5.3.3 EFFECTIVENESS OF PIXEL NECESSITY ESTIMATION

We further evaluate the effect of dynamically determining tool usage necessity in pixel necessity rollouts compared to using predefined necessity. The predefined necessity is obtained by running our SFT model with a temperature of 1.0 and collecting 8 rollouts per query (Pass@8); for each query, if the majority of rollouts involve tool usage, the necessity is set to "tool," otherwise to "no-tool". Figure 3 (a) shows the accuracy across five benchmarks. The predefined necessity approach achieves an average accuracy of 72.1, which is lower than Pixel Reasoner (Su et al., 2025a) and significantly below our adaptive method. This demonstrates that static necessity assignment cannot adapt to changes in the model's capability and thus fails to reliably estimate whether a query requires tool usage during the training process, leading to substantial accuracy loss. The performance gap is most pronounced on HR-Bench 4K/8K, where predefined necessity reduces the model's ability to handle high-resolution visual reasoning. Figure 3 (b) reports the ratio of tool usage across benchmarks.

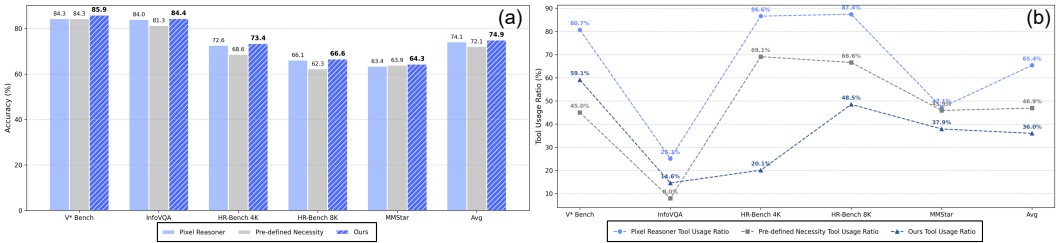

Figure 3: Ablation study on the effectiveness of pixel necessity estimation, showing benchmark accuracy (a) and tool usage ratio (b).

Although predefined necessity produces a tool ratio of 46.9, falling between Pixel Reasoner and our method, it fail to deliver the same accuracy improvements. This indicates that while predefined necessity reduces redundant pixel-level operations compared to Pixel Reasoner, they cannot match the flexibility of adaptive reasoning. Our adaptive strategy enables the model to make more informed decisions about when to invoke tools, improving both accuracy and efficient tool utilization.

### 5.3.4 EFFECTIVENESS OF REWARDS IN PIXEL NECESSITY ROLLOUTS

Table 4 evaluates the effectiveness of incorporating rewards from the pixel necessity rollouts during RGRL. When the rewards from the first eight rollouts (forced tool and forced no-tool) are excluded

from gradient updates (Ours w/o PN rewards), the model attains an average accuracy of 73.7 across the five benchmarks. Incorporating these rewards consistently improves performance, with our full method reaching 74.9 on average. These results confirm that the rewards in pixel necessity rollouts provide reliable tool necessity for learning when tool usage is truly beneficial, which subsequently enhances the adaptive rollouts.

Table 4: Ablation study on the effectiveness of rewards in the pixel necessity rollouts.

| Model | V* Bench | MMStar | HR-Bench 4K | HR-Bench 8K | InfoVQA | Avg |
|---|---|---|---|---|---|---|
| Ours w/o PN rewards | 85.3 (68.1) | 64.0 (54.9) | 73.1 (21.9) | 64.0 (39.0) | 82.1 (1.9) | 73.7 (37.2) |
| **Ours** | **85.9** (59.1) | **64.3** (37.9) | **73.4** (20.1) | **66.6** (48.5) | **84.4** (14.6) | **74.9** (36.0) |

## 5.4 CASE STUDY

Figure 4 illustrates two representative cases. On the left, for archaeological site sign text recognition, Pixel Reasoner conducts redundant cropping operations, introducing interfering visual information and thus failing to identify the correct text. In contrast, our model focuses on the key sign, clearly recognizing the text and outputting the correct answer "ISTRE.PULA" without unnecessary steps. On the right, for cricket statistics comparison, Pixel Reasoner makes multiple incorrect crops and miscalculates, while our model accurately locates the relevant statistics in the infographic and solves the task directly, yielding the correct answer "95". These cases show that our adaptive framework improves efficiency by avoiding unnecessary operations and enhances robustness by making more reliable tool-use decisions. For more examples, please refer to Appendix E.

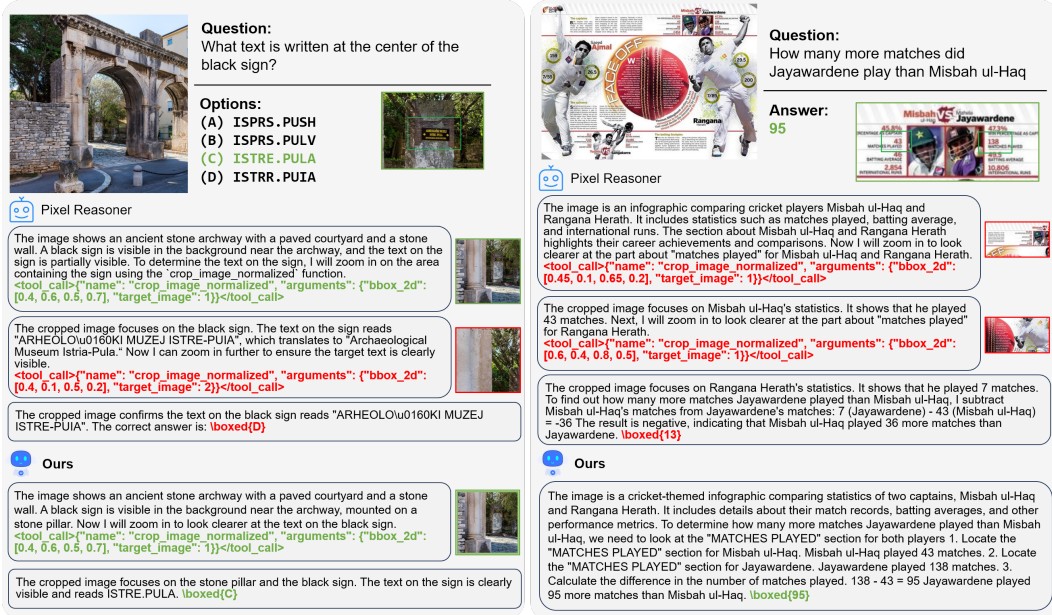

Figure 4: Comparison between Pixel Reasoner and our method on multimodal reasoning tasks. Left: Archaeological site sign text recognition. Right: Cricket statistics comparison.

## 6 CONCLUSION

In this work, we introduced a framework for adaptive pixel-space reasoning in multimodal reasoning tasks. By combining operation-aware supervised fine-tuning with rollout-guided reinforcement learning, the model learns query-specific strategies for deciding when to invoke pixel-level operations. Compared to other pipelining and end-to-end multimodal reasoning methods, the proposed approach demonstrates the ability to dynamically adapt to varying query complexities, avoiding both neglect and overuse of pixel-level operations. Extensive experiments across five benchmarks confirm that this framework consistently improves accuracy and efficiency, validating the effectiveness of our adaptive pixel-space reasoning framework.

ETHICS STATEMENT

Our work aims to enhance the adaptive pixel-space reasoning ability of VLMs without introducing any additional ethical concerns or resolving existing ones.

REPRODUCIBILITY STATEMENT

We propose a training framework for adaptive pixel-space reasoning. All reward formulations, roll-out configurations, and evaluation protocols are described in detail in the main paper. Specifically, Appendix B lists all prompts used in training, Appendix C provides the complete hyperparameters for both SFT and RL stages, and Appendix E includes additional case studies to facilitate further analysis and verification. Our code, data, and models will be publicly accessible.

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

# APPENDIX

## A  THE USE OF LARGE LANGUAGE MODELS (LLMS)

Large Language Models (LLMs) are employed solely to assist with language refinement and stylistic polishing of the manuscript. They are not involved in research ideation, experimental design and analysis. All conceptual and technical contributions are the sole responsibility of the authors.

## B  DETAILED PROMPTS

We provide the exact prompts used in different training phases. Specifically, the SFT stage uses the instruction template in Prompt 1, which aims to establish foundational competence in both pure textual CoT and the proper execution of visual operations. In the RL stage, the prompts for pixel necessity estimation rollouts enforce opposite behaviors: Prompt 2 explicitly instructs the model to invoke zoom-in, while Prompt 3 prohibits its use. These controlled settings enable the model to learn the correspondence between query type and tool necessity. In contrast, the adaptive rollout phase adopts the neutral prompt in Prompt 4, where the model is free to decide whether or not to perform pixel-space reasoning. This setup ensures that the model is first exposed to both extremes during necessity estimation and then given autonomy to balance textual reasoning and visual operations during adaptive rollouts.

## C  TRAINING HYPERPARAMETERS

Table A1 and A2 summarize the key hyperparameters for both the supervised fine-tuning (SFT) and reinforcement learning (RL) stages. The SFT phase initializes the model with baseline competence in pure textual CoT and pixel-space operations, specifying optimizer, learning rate schedule, batch sizes and frozen vision modules.

The RL phase trains the model for adaptive tool usage through rollout-guided reinforcement learning. Key settings include global and micro batch sizes, replay buffer size, number of samples and episodes, input/output lengths, learning rate, KL coefficient, train temperature, top-p sampling, reward and its coefficients.

## D  BENCHMARK DETAILS

We evaluate our method across five diverse multimodal benchmarks, covering both fine-grained perception and complex high-level reasoning. V* (V-Star) Bench (Wu & Xie, 2024) assesses the ability

---

**Prompt 1:** *SFT Prompt*

*You are a helpful assistant.*

*# Tools*

*You may call one or more functions to assist with the user query.*

*You are provided with function signatures within <tools></tools> XML tags: <tools>{"type": "function", "function": {"name": "crop_image_normalized", "description": "Zoom in on the image based on the bounding box coordinates. It is useful when the object or text in the image is too small to be seen.", "parameters": {"type": "object", "properties": {"bbox_2d": {"type": "array", "description": "coordinates for bounding box of the area you want to zoom in. Values should be within [0.0,1.0].", "items": {"type": "number"}}, "target_image": {"type": "number", "description": "The index of the image to crop. Index from 1 to the number of images. Choose 1 to operate on original image."}, "required": ["bbox_2d", "target_image"]}}} </tools>*

*For each function call, return a json object with function name and arguments within <tool_call></tool_call> XML tags: <tool_call>{"name": <function-name>, "arguments": <args-json-object>} </tool_call>*

*[image]*

*[question]*

*Guidelines: Understand the given visual information and the user query. Determine if it is beneficial to employ the given visual operations (tools). We can look closer by crop_image. Reason with the visual information step by step, and put your final answer within* `\boxed{}`*.*

---

**Prompt 2:** *RL Prompt for Tool Use in Pixel Necessity Rollouts*

*You are a helpful assistant.*

*# Tools*

*You may call one or more functions to assist with the user query.*

*You are provided with function signatures within <tools></tools> XML tags: <tools>{"type": "function", "function": {"name": "crop_image_normalized", "description": "Zoom in on the image based on the bounding box coordinates. It is useful when the object or text in the image is too small to be seen.", "parameters": {"type": "object", "properties": {"bbox_2d": {"type": "array", "description": "coordinates for bounding box of the area you want to zoom in. Values should be within [0.0,1.0].", "items": {"type": "number"}}, "target_image": {"type": "number", "description": "The index of the image to crop. Index from 1 to the number of images. Choose 1 to operate on original image."}, "required": ["bbox_2d", "target_image"]}}} </tools>*

*For each function call, return a json object with function name and arguments within <tool_call></tool_call> XML tags: <tool_call>{"name": <function-name>, "arguments": <args-json-object>} </tool_call>*

*[image]*

*[question]*

*Guidelines: Understand the given visual information and the user query. You must zoom in on the image using the tool (crop_image). Reason with the visual information step by step, and put your final answer within* `\boxed{}`*.*

---

of VLMs to handle visually intricate, high-resolution images and capture subtle details. MMStar (Chen et al., 2024) focuses on general-purpose multimodal reasoning, testing comprehension across a broad set of tasks involving textual and visual interactions. HR-Bench (Wang et al., 2024b) (HR-

---

**Prompt 3:** *RL Prompt for No Tool Use in Pixel Necessity Rollouts*

*You are a helpful assistant.*

*[image]*

*[question]*

*Guidelines: Understand the given visual information and the user query. Reason with the visual information step by step, and put your final answer within* `\boxed{}`.

---

**Prompt 4:** *RL Prompt for Adaptive Tool Use in Adaptive Rollouts*

*You are a helpful assistant.*

*# Tools*

*You may call one or more functions to assist with the user query.*

*You are provided with function signatures within <tools></tools> XML tags: <tools>{"type": "function", "function": {"name": "crop_image_normalized", "description": "Zoom in on the image based on the bounding box coordinates. It is useful when the object or text in the image is too small to be seen.", "parameters": {"type": "object", "properties": {"bbox_2d": {"type": "array", "description": "coordinates for bounding box of the area you want to zoom in. Values should be within [0.0,1.0].", "items": {"type": "number"}}, "target_image": "type": "number", "description": "The index of the image to crop. Index from 1 to the number of images. Choose 1 to operate on original image."}, "required": ["bbox_2d", "target_image"]}}} </tools>*

*For each function call, return a json object with function name and arguments within <tool_call></tool_call> XML tags: <tool_call>{"name": <function-name>, "arguments": <args-json-object>} </tool_call>*

*[image]*

*[question]*

*Guidelines: Understand the given visual information and the user query. Determine if it is beneficial to employ the given visual operations (tools). We can look closer by crop_image. Reason with the visual information step by step, and put your final answer within* `\boxed{}`.

---

Bench 4K/8K) are specifically designed to probe the capability of models in dealing with ultra-high-resolution images, where reasoning often requires identifying small-scale objects or subtle visual cues that are easily overlooked. Finally, InfographicVQA (InfoVQA) (Mathew et al., 2022) emphasizes reasoning over infographic-style images that tightly integrate diagrams, charts, and textual annotations, requiring precise alignment between textual information and visual layout.

# E  MORE CASES

To complement the main experiments, we provide additional qualitative comparisons in Figure A1–A5. These cases illustrate how our model adapts its tool usage across different scenarios. They serve as concrete examples to better understand the model's reasoning behaviors beyond aggregate metrics.

Figure A1 displays a case from the license plate recognition task, illustrating the reasoning processes of Pixel Reasoner and our method. The goal is to determine the license plate number of the vehicle in the image among the provided options. Pixel Reasoner makes multiple attempts at cropping, first focusing on irrelevant pavement areas before eventually finding the van and its license plate. Our method, however, efficiently zooms in on the van in a single cropping step, directly retrieving the

Table A1: SFT and RL hyperparameters.

| Parameter | Value |
|---|---|
| Number of nodes | 1 |
| GPUs per node | 8 |
| Total epochs | 5 |
| Seed | 49 |
| Optimizer | AdamW |
| Learning rate | $1.0 \times 10^{-6}$ |
| Scheduler | Cosine decay |
| Warmup ratio | 0.1 |
| Per-device batch size | 1 |
| Gradient accumulation steps | 2 |
| Precision | bfloat16 (BF16) |
| Gradient checkpointing | Enabled |
| Attention implementation | FlashAttention-2 |
| Freeze vision modules | True |

(a) SFT hyperparameters

| Parameter | Value |
|---|---|
| Training batch size (global) | 256 |
| Micro batch size (per actor) | 2 |
| Replay buffer size | 512 |
| Rollout batch size | 512 |
| Number of samples per prompt | 16 |
| Number of epochs | 3 |
| Max input length | 2048 |
| Max generation length | 10000 |
| Actor learning rate | $1.0 \times 10^{-6}$ |
| Zero Redundancy Stage | 3 |
| Auxiliary loss coefficient | 0.05 |
| KL coefficient | 0.0 |
| Train Temperature | 1.0 |
| Top-p | 0.95 |
| Precision | bfloat16 (BF16) |
| Gradient checkpointing | Enabled |
| Attention implementation | FlashAttention |

(b) RL hyperparameters

Table A2: RL reward and its coefficients.

| Category | Setting |
|---|---|
| Pixel Necessity | $b_1$: 1.2
$c_1$: 1.0
$\lambda_{\text{instr}}$: 0.08 |
| Adaptive | $b_2$: 1.6
$c_2$: 0.8
$b_3$: 1.2
$c_3$: 1.0
$\lambda_{\text{adapt}}$: 0.05 |
| Rollout Consistency | $\gamma$: 0.1 |

correct license plate number "V-223-LV". This case exemplifies how our approach optimizes tool utilization for more efficient and precise multimodal reasoning in the context of this case study.

Figure A2 presents a case from the task of determining the year Sachin Tendulkar reached the Guinness World Record for being the first player to score 10,000 runs, comparing the reasoning processes of Pixel Reasoner and our method. Pixel Reasoner attempts to zoom in on a section of the infographic's timeline but ends up with an incorrect year, 2005. Our method, on the other hand, directly analyzes the infographic's content and accurately identifies the correct year, 2001, without unnecessary tool-based cropping. This case demonstrates the effectiveness of our approach in efficiently and accurately reasoning about such sports-related milestone-finding tasks compared to Pixel Reasoner.

As shown in Figure A3, it presents a case from the task of identifying which two numbered locations on a provided map belong to the same country, comparing the reasoning processes of Pixel Reasoner and our method. Pixel Reasoner zooms in on the map and incorrectly concludes that locations 2 and 3 belong to the same country, selecting option B. Our method, through analyzing the geographical locations of each numbered marker, accurately determines that locations 1 and 2 are both in the United Kingdom, thus selecting the correct option C. This case illustrates how our approach excels in precise geographical reasoning and correct option selection compared to Pixel Reasoner in such map-based country association tasks, highlighting the latter's error in misidentifying the affiliation of location 3.

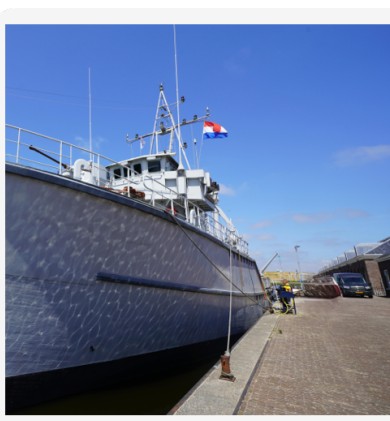

**Question:**
What is the license plate number of the vehicle in the image?

**Options:**
(A) V-223-LV
(B) V-322-LV
(C) V-224-LV
(D) V-223-LW

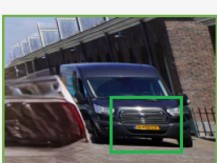

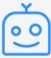 Pixel Reasoner

The image shows a docked ship with a paved area beside it. A van is parked near the ship, and its license plate is visible. The license plate appears to be partially obscured, but the visible portion matches the format of a Dutch license plate. Now I will zoom in to look clearer at the license plate.
**<tool_call>{"name": "crop_image_normalized", "arguments": {"bbox_2d": [0.75, 0.8, 0.85, 0.9], "target_image": 1}}</tool_call>**

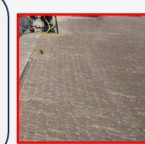

The cropped image focuses on the pavement area near the van. The license plate is not visible in this cropped section, indicating that the target image might need further zooming or adjusting. Now I will zoom in again to get a clearer view of the license plate.
**<tool_call>{"name": "crop_image_normalized", "arguments": {"bbox_2d": [0.8, 0.7, 0.9, 0.8], "target_image": 1}}</tool_call>**

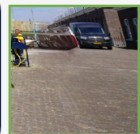

The cropped image clearly shows the van and its license plate. The license plate reads **V-223-LV**. **\\boxed{A}**

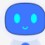 **Ours**

The image shows a docked boat with a flag and a paved area. In the background, there is a van parked near a building. The license plate of the van is visible, and the task is to identify its number. Now I will zoom in to look clearer at the license plate of the van.
**<tool_call>{"name": "crop_image_normalized", "arguments": {"bbox_2d": [0.8, 0.7, 0.9, 0.8], "target_image": 1}}</tool_call>**

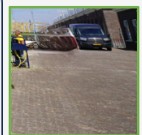

The cropped image focuses on the van. The license plate clearly displays the number **V-223-LV**. **\\boxed{A}**

Figure A1: Case of license plate recognition task comparison between Pixel Reasoner and our method.

As shown in Figure A4, it presents a case from the task of determining how many types of bonding exist in chemistry, comparing the reasoning processes of Pixel Reasoner and our method. Pixel Reasoner zooms in on a section of the "Map of Chemistry" but only identifies two types of bonding (covalent and ionic), leading to an incorrect answer of 2. Our method, by strategically locating the "BONDS" section in the lower-left part of the infographic, accurately identifies four types of bonding: Covalent Bond, Ionic Bond, van der Waals bonding, and Hydrogen Bond, thus obtaining the correct answer of 4. This case demonstrates how our approach enables more comprehensive and accurate information retrieval in chemical concept-related reasoning tasks compared to Pixel Reasoner.

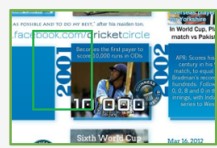

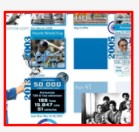

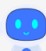

**Question:**
Which year did he reach the guinness world record for becoming first player to reach 10K runs

**Answer:**
**2001**

**Pixel Reasoner**

The image is an infographic about Sachin Tendulkar, detailing his career highlights, achievements, and personal milestones. It includes timelines, statistics, and quotes about his cricket career. Now I will zoom in to look clearer at the part about "the year he reached the Guinness World Record for becoming the first player to reach 10K runs.
**<tool_call>{"name": "crop_image_normalized", "arguments": {"bbox_2d": [0.16, 0.6, 0.22, 0.7], "target_image": 1}}</tool_call>**

The cropped image focuses on the timeline of Tendulkar's career. The section mentions that he became the first player to score 10,000 runs in ODIs on October 5, 2012. **\boxed{2005}**

**Ours**

The image is an infographic about Sachin Tendulkar, detailing his career highlights, achievements, and personal milestones. It includes timelines, statistics, and quotes about his cricket career. To determine the year he reached the Guinness World Record for being the first player to reach 10,000 runs, I'll look for a specific section in the infographic that mentions this milestone. **\boxed{2001}**

Figure A2: Case of Sachin Tendulkar's Guinness World Record year determination task comparison between Pixel Reasoner and our method.

For coronavirus-related geographic reasoning in Figure A5, Pixel Reasoner engages in two crop attempts yet fails to zero in on the correct location each time. This points to a tendency of invoking the cropping tool in a mechanical, almost obligatory manner—"calling the tool just for the sake of tool invocation"—without a strategic, solution-oriented assessment of when and where to crop. In contrast, our model accurately targets the key geographic information in a more direct way and produces the correct answer "Thailand" without redundant operations.

Through the five case studies (license plate recognition, Sachin Tendulkar's record year determination, same-country location identification on a map, chemical bonding type counting and coronavirus-related geographic reasoning), we observe the strengths of our method in adaptive pixel-space reasoning.

In each case, baselines like Pixel Reasoner either overused pixel-level operations (e.g., redundant cropping in license plate recognition and map tasks, leading to inefficiency or errors) or failed to invoke visual inspection when necessary (missing critical visual cues, as seen in the chemical bonding task where Pixel Reasoner identified only partial bonding types).

In contrast, our model adaptively decides when to perform fine-grained visual operations (e.g., targeted zoom-in for license plate recognition, direct content analysis for cricket statistics and chemical bonding) or rely on semantic reasoning. By combining operation-aware supervised fine-tuning and rollout-guided reinforcement learning, it balances the need for pixel-level operations and high-level reasoning: it avoids overusing compute-intensive pixel operations while capturing critical visual details. This adaptive strategy, guided by rewards for correctness, instruction-following, adaptive tool-

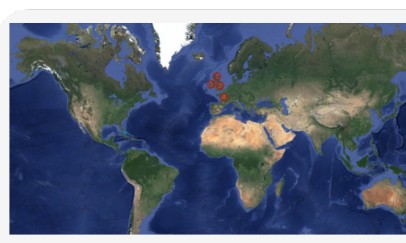 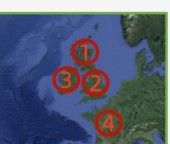

**Question:**
On the provided map, which two numbered locations belong to the same country?

**Options:**
(A) Locations 2 and 4
(B) Locations 2 and 3
(C) Locations 1 and 2
(D) Locations 1 and 3

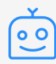 Pixel Reasoner

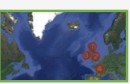

The image is a world map highlighting specific regions with numbered markers. The numbered locations are positioned in Europe, with some overlapping or close proximity. Now I will zoom in to look clearer at the locations numbered 1, 2, 3, and 4.
**<tool_call>{"name": "crop_image_normalized", "arguments": {"bbox_2d": [0.4, 0.2, 0.5, 0.3], "target_image": 1}}</tool_call>**

The cropped image focuses on the northern part of Europe, specifically the British Isles. Locations 2 and 3 are marked on the British Isles, which are part of the same country, the United Kingdom. **\boxed{B}**

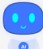 **Ours**

The image is a world map highlighting specific regions with numbered markers. The numbers (1, 2, 3, 4) are placed over geographical locations, likely representing significant points or countries. To determine which two locations belong to the same country, I need to identify the countries associated with each numbered location: Location 1 is in the northern part of Europe, specifically in the United Kingdom. Location 2 is also in the northern part of Europe, specifically in the United Kingdom. Location 3 is in the northern part of Europe, specifically in Norway. Location 4 is in the northern part of Europe, specifically in Denmark. Both locations 1 and 2 are in the United Kingdom, which explains their proximity and shared characteristics. **\boxed{C}**

Figure A3: Case of identifying same-country locations on a map: Comparison between Pixel Reasoner and our method.

necessity alignment, and rollout consistency, achieves accurate results across diverse multimodal reasoning tasks—from visual identification to knowledge-based querying—surpassing both general VLMs and tool-augmented baselines, and validating the effectiveness of adaptive pixel-space reasoning.

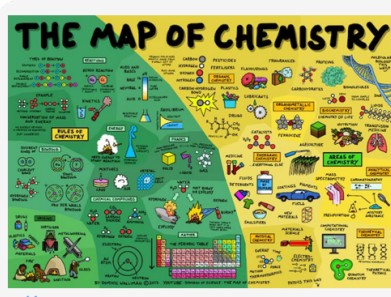

**Question:**

How many types of bonding are there in chemistry?

**Answer:**

**4**

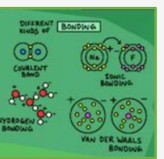

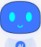 Pixel Reasoner

The image is a colorful and detailed "Map of Chemistry" that illustrates various concepts, rules, and areas of chemistry. It includes sections on types of reactions, rules of chemistry, bonding, and different areas of chemistry. Now I will zoom in to look clearer at the part about "types of bonding."
**<tool_call>{"name": "crop_image_normalized", "arguments": {"bbox_2d": [0.01, 0.4, 0.2, 0.5], "target_image": 1}}</tool_call>**

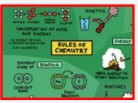

The cropped image focuses on "Rules of Chemistry," specifically highlighting "Different kinds of Bonding." It lists two types of bonding: covalent and ionic. **\boxed{2}**

**Ours**

The image is a colorful and detailed "Map of Chemistry" that illustrates various concepts, rules, and areas of chemistry. It includes sections on chemical reactions, types of bonds, the periodic table, and different areas of chemistry. To determine the number of types of bonding in chemistry, I'll look for the section labeled "BONDS" in the image. 1. Locate the "BONDS" section in the image. 2. Identify the types of bonds listed there. The "BONDS" section is located in the lower-left part of the image. It lists the following types of bonds: Covalent Bond, Ionic Bond, van der Waals bonding, Hydrogen Bond. There are four types of bonds listed in this section. **\boxed{4}**

Figure A4: Case of determining the number of chemical bonding types: Comparison between Pixel Reasoner and our method.

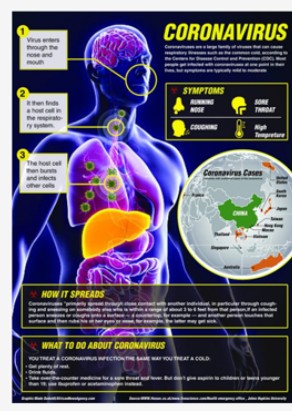
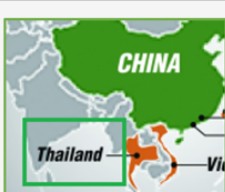

**Question:**

Which country is shown in this infographic, right below china, on the left side?

**Answer:**
thailand

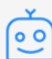 Pixel Reasoner

The document is an infographic about the coronavirus, detailing its spread, symptoms, and treatment. It includes a section on how the virus spreads, listing countries with confirmed cases. Now I will zoom in to look clearer at the part about "the country shown in this infographic, right below China, on the left side."
<tool_call>{"name": "crop_image_normalized", "arguments": {"bbox_2d": [0.4, 0.6, 0.6, 0.7], "target_image": 1}}</tool_call>

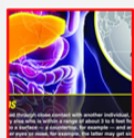

The cropped image focuses on the section about "How it Spreads," but the target country is not visible here. I will zoom in further to locate the target country.
<tool_call>{"name": "crop_image_normalized", "arguments": {"bbox_2d": [0.4, 0.7, 0.99, 0.8], "target_image": 1}}</tool_call>

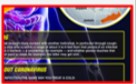

The cropped image clearly shows the map of countries with confirmed coronavirus cases. Directly below China, on the left side, is the country Thailand. \boxed{Thailand}

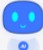 **Ours**

The document is an infographic about the coronavirus, detailing its spread, symptoms, and treatment. It includes a section on how the virus spreads, listing countries affected by the virus. The question asks for the country immediately below China on the left side of the infographic.
Key Observations: 1. The infographic includes a map of the world with highlighted countries affected by the coronavirus. 2. The map is positioned to the right of the central illustration of the human body. 3. The countries are listed in a clockwise direction from China.
Step-by-Step Reasoning: 1. Locate China on the map. 2. Identify the country directly below China. 3. The map shows Thailand positioned directly below China. \boxed{Thailand}

Figure A5: Case of coronavirus-related geographic reasoning: Comparison between Pixel Reasoner and our method.

