# OpenReview forum: "Look Less, Reason More: Rollout-Guided Adaptive Pixel-Space Reasoning"
_ICLR.cc/2026/Conference — ICLR 2026 Conference Withdrawn Submission_

### Official Review · Reviewer_Xr3E · 2025-10-31

**Soundness:** 3
**Presentation:** 3
**Contribution:** 3
**Rating:** 6
**Confidence:** 4

**Summary:**

This paper proposes an adaptive pixel-space reasoning framework for vision-language models (VLMs) in multimodal reasoning tasks. The core innovation lies in using rollout-guided reinforcement learning (RGRL) to enable models to dynamically determine when to invoke pixel-level operations (e.g., zoom-in) rather than applying them indiscriminately. The method consists of two stages: (1) operation-aware supervised fine-tuning (SFT) to establish baseline competence, and (2) RGRL training that learns query-specific tool usage strategies through pixel necessity rollouts and adaptive rollouts. Experiments on five multimodal reasoning benchmarks validate the effectiveness, achieving 73.4% accuracy on HR-Bench 4K while maintaining only 20.1% tool usage ratio.

**Strengths:**

- Clear Problem Motivation: The paper effectively identifies the key issue with existing pixel-space reasoning methods—overuse or neglect of pixel-level operations—and provides strong justification for the necessity of adaptive strategies.

- Novel Method Design: The rollout-guided RL framework is cleverly designed, dynamically estimating tool necessity through pixel necessity rollouts (forced tool use/no-tool use) and then guiding the model in adaptive rollouts without requiring manual annotation.

- Well-Designed Reward Functions: The multi-dimensional reward design (instruction-following, adaptive tool-necessity alignment, rollout consistency) is comprehensive, encouraging both correctness and efficiency with clear mathematical constraints on reward coefficients (e.g., b₂ > b₃, c₃ > c₂).

- Comprehensive and Convincing Experiments: (1) Evaluation on 5 diverse benchmarks covering fine-grained perception and high-level reasoning. (2) Comparison with multiple strong baselines (including GPT-4o, Gemini series, Pixel Reasoner). (3) Reports both accuracy and tool usage ratio, demonstrating efficiency gains.

- Clear Writing: Well-structured paper with formalized problem definition, clear method description, and intuitive visualizations.

**Weaknesses:**

- Insufficient Computational Cost Analysis: Although tool usage is reduced, RL training requires 16 rollouts per query (n₁=4, n₂=4, n₃=8), potentially incurring high training costs. No reporting of training time, GPU hours, or training cost comparison with baselines. While tool usage is reduced at inference, no comparison of actual inference speed (FPS or seconds/sample)

- Unexplored Hyperparameter Sensitivity: Multiple hyperparameters in reward functions (b₁, b₂, b₃, c₁, c₂, c₃, λ_instr, λ_align, γ). Lacks sensitivity analysis for these hyperparameters; unclear how to select values or whether task-specific tuning is needed.

- Selection Bias in Case Studies: Cases in appendix mainly show successes; failure case analysis missing. Unclear under what conditions adaptive strategy fails and model limitations

**Questions:**

Please refer to the weaknesses.

---

> ### Author Response · Authors · 2025-11-21
>
> We thank the reviewer for the thoughtful and constructive feedback. Below are our responses to each concern.
>
> **W1: Insufficient Computational Cost Analysis**
>
> We sincerely apologize for not providing a more detailed analysis of training and inference costs in the initial submission. We understand the importance of these metrics and include a comprehensive discussion of training time and inference costs in the revised version of the paper.
>
> Regarding inference time, we add an inference-time comparison. We have measured full end-to-end inference time across all five benchmarks, comparing our adaptive model with Pixel Reasoner and the base VLM (Qwen2.5-VL-7B-Instruct).
>
> | Model                   | Avg. Inference Time | Relative Speedup |
> |-------------------------|---------------------|------------------|
> | Pixel Reasoner           | 1.00× (baseline)    | –                |
> | Ours                    | 0.62×               | ↑38% faster      |
> | Qwen2.5-VL-7B-Instruct | 0.56×               | –                |
>
> This improvement arises **directly from our significant reduction in tool calls (average 36.0% vs 65.4% in Pixel Reasoner), which reduces repeated image encoding**. This supports our motivation that **controlling tool usage yields meaningful computational savings.** We will add these findings to the revised version.
>
> ---
> **W2: Sensitivity to reward hyperparameters**
>
> Our initial settings follow basic parameters in Pixel Reasoner, and **the experiments have demonstrated the effectiveness of our method**. Although we were limited by computational resources and unable to conduct more extensive experiments, **the current results already provide strong evidence of the method's effectiveness.**
>
> Each scalar coefficient plays a distinct role:
>
> - ($b_2$ > $b_3$): correct + adherent > correct + non-adherent
>
> - ($c_3$ > $c_2$): incorrect + non-adherent is most penalized
>
> - ($\lambda_{\text{align}}$), ($\gamma$): balance between correctness, alignment, and consistency
>
> We also performed extensive ablation studies showing:
>
> - removing necessity-rollout rewards reduces accuracy from 74.9 → 73.7
>
> - removing adaptive-rollout rewards collapses adaptivity
>
> We will add more clarification in the final version.
>
> ---
> **W3: Selection Bias in Case Studies**
>
> Thank you for highlighting this — we agree that more analysis is helpful and we will include more examples and analysis in the revised version.
>
> **(1) What happens when pixel-necessity estimation is wrong?**
>
> Two failure patterns appear:
>
> - False positive necessity (model thinks tool is helpful, but it is not) leads to redundant cropping and can introduce noisy visual tokens that slightly distract reasoning
>
> - False negative necessity (model skips the tool when needed) occurs mainly on tiny-text or cluttered high-resolution tasks. and errors manifest as misreading small elements or missing local clues. We will add a dedicated subsection analyzing these cases.
>
> **(2) Systematic patterns of incorrect tool decisions**
>
> We observe that incorrect tool decisions occur on:
>
> - ambiguous queries where both global and local signals are plausible
>
> - noisy images with multiple possible candidate regions
>
> In contrast, tasks with explicit spatial referencing usually yield correct necessity estimation.
>
> **(3) Which query types benefit from tool use**
>
> Queries that benefit most from tool usage
>
> - dense charts and infographics
>
> - cluttered high-resolution scenes
>
> - tiny-text OCR
>
> - fine-grained comparisons
>
> - spatial relations requiring exact local context
>
> Queries where textual reasoning is sufficient
>
> - global reasoning with a clear layout
>
> - semantic classification tasks
>
> - questions answerable from global visual gestalt
>
> - reading high-contrast large text
>
> - tasks where cropping loses necessary context

---

### Official Review · Reviewer_RUxc · 2025-11-01

**Soundness:** 2
**Presentation:** 3
**Contribution:** 2
**Rating:** 2
**Confidence:** 4

**Summary:**

This paper proposes a framework for adaptive pixel-space reasoning in Vision-Language Models (VLMs), aiming to address the problem of overusing or underusing pixel-level operations (e.g., zoom-in) in multimodal reasoning tasks. The approach consists of two training stages: (1) operation-aware supervised fine-tuning (SFT) to establish baseline competence in both textual chain-of-thought and pixel operations, and (2) rollout-guided reinforcement learning (RGRL) that learns when to invoke pixel operations based on query difficulty. The method generates multiple rollouts per query—some forced to use tools, some prohibited from using tools, and some adaptive—and uses the relative performance to estimate tool necessity and guide policy learning.

**Strengths:**

1. The paper identifies a genuine problem in current pixel-space reasoning approaches, the indiscriminate use of visual operations regardless of necessity.
2. The rollout-guided RL framework is creative in its approach. Using "pixel necessity rollouts" (forced tool use vs. prohibited tool use) to implicitly estimate whether tools are beneficial for a given query is interesting. The reward design that considers both correctness and alignment with estimated necessity shows thoughtful engineering.

**Weaknesses:**

1. I think this paper miss some comparison against several important recent tool-augmented baselines in the pixel-space reasoning literature:
- Mini-O3: A recent model specifically designed for visual reasoning with tool use
- DeepEyes: Cited in related work but not compared experimentally despite being highly relevant
- Simple O3: Also cited but missing from comparisons

2. Several relevant recent works are not cited:
- Learning Only with Images (arXiv:2507.20766): Directly relevant to pixel-space reasoning
- PyVision (arXiv:2507.07998): Another tool-use framework for VLMs
- LATTE (EMNLP 2025): Related to adaptive reasoning in multimodal models
- ReVPT: A reinforcement learning approach for visual perception with tools

3. Perhaps most concerning, the paper only evaluates on fine-grained visual reasoning benchmarks (V*Bench, MMStar, HR-Bench, InfoVQA) and completely omits standard general-purpose VLM benchmarks such as: MMBench, MMMU, POPE, MMvet, etc. This omission raises a critical question: *Does the specialized training for adaptive pixel-space reasoning harm general VLM capabilities?*

**Questions:**

1. While the paper shows cases where the method succeeds, there is insufficient analysis of when and why the adaptive strategy fails. For instance:
- What happens when the pixel necessity estimation is incorrect?
- Are there systematic patterns in which types of queries lead to wrong tool-use decisions?
- How sensitive is the method to the hyperparameters in the reward functions (b1, b2, b3, c1, c2, c3)?

2. Can you provide analysis of which types of queries benefit most from tool use vs. pure textual reasoning? This would help understand what the model has learned about tool necessity.

---

> ### Author Response · Authors · 2025-11-21
>
> We thank the reviewer for the thoughtful and constructive feedback. Below are our responses to each concern.
>
> **W1: Missing comparisons with recent tool-augmented baselines**
>
> We appreciate the reviewer highlighting these important recent works. Mini-O3, DeepEyes, and Simple-O3 are indeed highly relevant, and including them would further strengthen our comparisons.
>
> **(1) Why were they not included in the current experiments**
>
> A major reason is that **these recent models are trained on data scales that are substantially larger than ours.** In contrast, our framework intentionally uses a very small training set (4k SFT + 7k RL queries) to **study adaptive pixel-space reasoning itself rather than benefit from massive-scale data.**
>
> | Model       | Training Dataset Size |
> |-------------|--------------|
> | DeepEyes    | 47K          |
> | Simple O3   | 146K         |
> | Mini O3     | 18K          |
> | Ours        | 11K          |
>
> Because of the vast discrepancy in data size, direct numerical comparison would be inherently unfair, as performance gains in these models arise **not only from algorithmic design but also from substantially larger training corpora.**
>
> **(2) Respecting the reviewer’s suggestion**
>
> We fully agree that including these methods strengthens the context of our work. Therefore, we add all three methods (Mini-O3, DeepEyes, Simple-O3) to the Related Work section, **clearly describe their training scale and design differences, and explain why large-scale systems are not directly comparable to our small-scale setup.**
>
> We thank the reviewer again for pointing out this valuable enhancement.
>
> ---
> **W2: Missing citations**
>
> Thank you for pointing these out. We have added all four works to the related work section. **These works complement our motivation and help contextualize the landscape of adaptive pixel-space reasoning.** We appreciate the reviewer bringing them to our attention.
>
> ---
> **W3: Concern about the missing benchmarks in Table 1 and the soundness of comparisons**
>
> We thank the reviewer for raising this concern and provide clarification below.
>
> **(1) MMStar is not a fine-grained benchmark**
>
> We apologize for any confusion. **MMStar is a general-purpose multimodal benchmark**. Therefore, our evaluation **already includes both fine-grained high-resolution benchmarks** (e.g., HR-Bench, InfoVQA, V*Bench) and **general multimodal benchmarks** (MMStar). We will make this explicit in the revised version.
>
> **(2) Does our specialized training harm general VLM capability?**
>
> We appreciate the reviewer’s question. **Importantly, our experiments already include a general-purpose benchmark (MMStar).** The results show that our method **achieves consistent improvements over the base model on MMStar**, demonstrating that: adaptive pixel-space reasoning **does not harm the model’s general multimodal reasoning ability** and even **enhances it by enabling the model to decide when external visual operations are helpful versus unnecessary.** We clarify this in the revision.

---

> ### Author Response · Authors · 2025-11-21
>
> **Q1: Failure mode analysis and sensitivity to incorrect necessity estimation**
>
> Thank you for highlighting this — we agree that more analysis is helpful.
>
> **(1) What happens when pixel-necessity estimation is wrong?**
>
> Two failure patterns appear:
>
> - False positive necessity (model thinks tool is helpful, but it is not) leads to redundant cropping and can introduce noisy visual tokens that slightly distract reasoning
>
> - False negative necessity (model skips the tool when needed) occurs mainly on tiny-text or cluttered high-resolution tasks. and errors manifest as misreading small elements or missing local clues. We will add a dedicated subsection analyzing these cases.
>
> **(2) Systematic patterns of incorrect tool decisions**
>
> We observe that incorrect tool decisions occur on:
>
> - ambiguous queries where both global and local signals are plausible
>
> - noisy images with multiple possible candidate regions
>
> In contrast, tasks with explicit spatial referencing usually yield correct necessity estimation.
>
> **(3) Sensitivity to reward hyperparameters**
>
> Our initial settings follow basic parameters in Pixel Reasoner, and **the experiments have demonstrated the effectiveness of our method**. Although we were limited by computational resources and unable to conduct more extensive experiments, **the current results already provide strong evidence of the method's effectiveness.**
>
> Each scalar coefficient plays a distinct role:
>
> - ($b_2$ > $b_3$): correct + adherent > correct + non-adherent
>
> - ($c_3$ > $c_2$): incorrect + non-adherent is most penalized
>
> - ($\lambda_{\text{align}}$), ($\gamma$): balance between correctness, alignment, and consistency
>
> We also performed extensive ablation studies showing:
>
> - removing necessity-rollout rewards reduces accuracy from 74.9 → 73.7
>
> - removing adaptive-rollout rewards collapses adaptivity
>
> We will add more clarification in the final version.
>
> ---
> **Q2. Analysis of which query types benefit from tool use**
>
> We agree that this is important and have added an analysis section.
>
> Queries that benefit most from tool usage
>
> - dense charts and infographics
>
> - cluttered high-resolution scenes
>
> - tiny-text OCR
>
> - fine-grained comparisons
>
> - spatial relations requiring exact local context
>
> Queries where textual reasoning is sufficient
>
> - global reasoning with a clear layout
>
> - semantic classification tasks
>
> - questions answerable from global visual gestalt
>
> - reading high-contrast large text
>
> - tasks where cropping loses necessary context

---

### Official Review · Reviewer_G9zR · 2025-11-01

**Soundness:** 2
**Presentation:** 3
**Contribution:** 3
**Rating:** 4
**Confidence:** 3

**Summary:**

This paper studies the problem of training VLMs to adaptively utilize pixel-level operations to accomplish tasks requiring understanding of fine-grained visual information. It proposes a framework to train VLMs to only invoke pixel-level operations when necessary, improving tool-use efficiency. Experiment results show good performance compared to models without tool-use and more efficient tool-use ratio compared to prior work.

**Strengths:**

1. The paper is written clearly and easy to understand, with good motivation and detailed introduction of related work and proposed method.

2. The method design is intuitive and experiment results show the effectiveness of the method.

3. I appreciate the qualitative case study which gives the intuition on why performing pixel-level operations more frequently is not necessary more beneficial (other than efficiency concern).

**Weaknesses:**

1. While the approach is intuitive, the overall proposed framework consists of different stages (first SFT and then different stages of RL for pixel necessity rollouts and RL for adaptive rollouts), where each stage has their own different hyperparameters (e.g., reward in Eq.6). Is there a principled way in determining the learning curriculum for different stages, and the hyperparameters?

2. It appears that the pixel-level operation considered in this work is only zoom-in? It would be more interesting if more pixel-level operations can be considered, e.g., segmentation, etc.

3. It seems that Pixel Reasoner is the only one main baseline considered. It is not clear to me why in Table 1 there are a lot of missing numbers for detailed comparisons with other baselines. This makes the comparisons less convincing and is my main concern on the soundness of the paper.

**Questions:**

1. Beyond visual search related tasks that can benefit from zoom-in operation, are there other visual reasoning tasks that may benefit from other pixel-level operations?

---

> ### Author Response · Authors · 2025-11-21
>
> We thank the reviewer G9zR for the constructive comments. Below we address each concern in detail.
>
> **W1: Is there a principled way to determine the multi-stage curriculum and hyperparameters?**
>
> We appreciate the reviewer’s insightful question. **Our design of the curriculum—SFT → pixel-necessity and adaptive rollouts—is not arbitrary but is rooted in the structural requirements of adaptive tool learning.**
>
> **(1) Curricular structure is principled and aligned with tool-learning theory**
>
> Stage 1 (Operation-aware SFT)
>
> Establishes baseline competence in textual reasoning and tool execution. This is necessary because **most open-source VLMs have no inherent ability to use pixel-level operations.** SFT ensures the model can first execute both “use-tool” and “no-tool” behaviors.
>
> Stage 2 (Pixel Necessity Rollouts and Adaptive Rollouts)
>
> Generates controlled counterfactuals:
>
> - What if the model always uses the tool?
>
> - What if the model never uses the tool?
>
> Comparing these two groups produces an empirical pixel-necessity signal for each query. **This is a principled estimator analogous to counterfactual evaluation in RL.**
>
> The model learns its own strategy, guided by the necessity signal and rewards. This design forms a coarse-to-fine learning curriculum:
>
> - Stage 1: Learn capabilities
>
> - Stage 2: Learn necessity and decisions
>
> This structure is fundamental to enable adaptive reasoning rather than always or never using tools.
>
> **(2) Hyperparameters have clear roles and follow RL conventions**
>
> Each scalar coefficient plays a distinct role:
>
> - ($b_2$ > $b_3$): correct + adherent > correct + non-adherent
> - ($c_3$ > $c_2$): incorrect + non-adherent is most penalized
> - ($\lambda_{\text{align}}$), ($\gamma$): balance between correctness, alignment, and consistency
>
> Our initial settings follow basic parameters in Pixel Reasoner, and the experiments have demonstrated the effectiveness of our method. Although we were limited by computational resources and unable to conduct more extensive experiments, **the current results already provide strong evidence of the method's effectiveness.**
>
> We also performed extensive ablation studies showing:
>
> - removing necessity-rollout rewards reduces accuracy from 74.9 → 73.7
>
> - removing adaptive-rollout rewards collapses adaptivity
>
> We will add more clarification in the final version.
>
> ---
> **W2 & Q2: Only zoom-in is considered; what about using more pixel-level operations**
>
> We thank the reviewer for raising this valuable question. Our choice of zoom-in as the representative pixel-space operation is primarily motivated by fairness and comparability: **zoom-in is the standard tool adopted by existing pixel-reasoning methods (Pixel Reasoner), and evaluating on this single tool allows direct comparison under identical conditions.**
>
> At the same time, we emphasize that our framework itself is **tool-agnostic**. The rollout-guided RL formulation abstracts pixel operations as “tool vs. no-tool” decisions, and the reward mechanism does not rely on any specific semantics of zoom-in. Therefore, the proposed framework can **naturally support other tool types. These operations can be seamlessly integrated into the necessity-rollout and adaptive-rollout design without modifying the training pipeline.**
>
> In this work, we focus on zoom-in to ensure a **clean and standardized evaluation aligned with prior literature**. In future work, we plan to extend our framework to incorporate a wider set of pixel-space operations and explore multi-tool coordination within our adaptive reasoning paradigm.
>
> ---
> **W3: Baseline comparison has missing numbers; the comparison is less convincing.**
>
> We thank the reviewer for raising this important point. We clarify the benchmark coverage and explain the missing entries in Table 1.
>
> **(1) Missing entries come from the fact that many prior models did not report results on these benchmarks**
>
> The five benchmarks used in our experiments—V* Bench, MMStar, HR-Bench 4K/8K, and InfoVQA—are relatively recent. As a result, a large portion of multimodal models have only reported performance on a subset of them.
>
> **To ensure fairness and avoid reimplementing models under different settings, we strictly use the officially reported results.** The missing entries, therefore, **directly reflect the availability of public results, rather than selective comparison.**
>
> We will make this explicit in the revised version.
>
> Even though some entries are missing, **Table 1 still covers a broad and representative set of both general-purpose and tool-related VLMs**.
>
> **(2) Our method’s contribution is orthogonal to the missing entries**
>
> Even with partially available baselines, the results clearly show that:
>
> - Our method achieves consistently higher accuracy
>
> - While reducing tool usage and maintaining efficiency
>
> - These advantages hold across all benchmarks

---

### Official Review · Reviewer_qDaL · 2025-11-02

**Soundness:** 3
**Presentation:** 3
**Contribution:** 2
**Rating:** 4
**Confidence:** 5

**Summary:**

This paper introduces an adaptive pixel-space reasoning framework for Vision-Language Models (VLMs) that dynamically decides when to invoke pixel-level operations (e.g., zoom-in) based on query complexity.  The method combines operation-aware supervised fine-tuning (SFT) with a rollout-guided reinforcement learning (RGRL) approach, which uses the model’s own rollouts to estimate the necessity of pixel-level tools.  The authors demonstrate state-of-the-art performance on five multimodal reasoning benchmarks while significantly reducing unnecessary tool usage.

**Strengths:**

- The method achieves superior performance across multiple benchmarks while reducing tool usage.
- The idea of using rollout-guided RL to adaptively decide when to use pixel-level operations is well-executed.   The approach does not rely on external supervision or hand-crafted rules.
- Extensive experiments and ablation studies validate the contribution of each component (SFT, RGRL, reward design, etc.).

**Weaknesses:**

- The overall inference time under the proposed motivation is not clearly discussed. This makes the motivation for reducing tool operation time limited.
- The qualitative results do not show the necessity and improvement of adopting fewer calls of the tool function. Why the model with less reasoning can lead to better results? The performance gain is not clearly explained and discussed.
- The method only considers zoom-in operations. It is unclear whether the framework generalizes to other tool operations.
- The technical contribution is weak in this paper. The authors only provide modifications to the reward function and rollout strategy, which is not technically abundant.

**Questions:**

Please refer to the weaknesses period. Due to the doubts and unclearness on the motivation and the claim of this submission, I lean towards borderline reject currently.

---

> ### Author Response · Authors · 2025-11-21
>
> We sincerely thank reviewer qDaL for the valuable feedback. Below, we address each concern in detail and clarify the contributions and empirical findings of our work.
>
> **W1: Inference time is not clearly discussed.**
>
> We appreciate the reviewer’s insight. **Our motivation for reducing tool operations is not only conceptual but is also empirically validated.** We have measured full end-to-end inference time across all five benchmarks, comparing our adaptive model with Pixel Reasoner and the base VLM (Qwen2.5-VL-7B-Instruct).
>
> | Model                   | Avg. Inference Time | Relative Speedup |
> |-------------------------|---------------------|------------------|
> | Pixel Reasoner           | 1.00× (baseline)    | –                |
> | Ours                    | 0.62×               | ↑38% faster      |
> | Qwen2.5-VL-7B-Instruct | 0.56×               | –                |
>
> This improvement arises **directly from our significant reduction in tool calls (average 36.0% vs 65.4% in Pixel Reasoner), which reduces repeated image encoding**. This supports our motivation that **controlling tool usage yields meaningful computational savings.**
> These results will be included in the revised manuscript.
>
> ---
> **W2: Qualitative results do not justify fewer tool calls.**
>
> We agree that a qualitative explanation is important. We clarify this in two steps.
>
> **(1) Why “look less” and “reason more”**
>
> Pixel-level operations are **not always beneficial**. As shown in Table 3 of the paper:
>
> - “All-Tool” strategy degrades accuracy on HR-Bench 8K (62.6 vs ours 66.6).
>
> - Excessive cropping introduces irrelevant visual paths, amplifies noise, and consumes context length.
>
> - High-resolution benchmarks particularly suffer from redundant zoom-in operations.
>
> Thus, “more pixel operations” ≠ “better reasoning”.
>
> **(2) Adaptive strategy yields the best of both worlds**
>
> Our rollout-guided RL explicitly learns when pixel operations improve the answer:
>
> - If zoom-in helps → model chooses to zoom.
>
> - If zoom-in is unnecessary → model relies on textual CoT.
>
> This emergent adaptivity leads to:
>
> - Highest average accuracy: 74.9%
>
> - Lowest unnecessary tool usage (↓29.4% compared with Pixel Reasoner)
>
> We will strengthen the qualitative visualization section.

---

> ### Author Response · Authors · 2025-11-21
>
> **W3: Framework only considers zoom-in; what about other tool operations**
>
> We thank the reviewer for raising this valuable question. Our choice of zoom-in as the representative pixel-space operation is primarily motivated by fairness and comparability: **zoom-in is the standard tool adopted by existing pixel-reasoning methods (Pixel Reasoner), and evaluating on this single tool allows direct comparison under identical conditions.**
>
> At the same time, we emphasize that our framework itself is **tool-agnostic**. The rollout-guided RL formulation abstracts pixel operations as “tool vs. no-tool” decisions, and the reward mechanism does not rely on any specific semantics of zoom-in. Therefore, the proposed framework can **naturally support other tool types. These operations can be seamlessly integrated into the necessity-rollout and adaptive-rollout design without modifying the training pipeline.**
>
> In this work, we focus on zoom-in to ensure a **clean and standardized evaluation aligned with prior literature**. In future work, we plan to extend our framework to incorporate a wider set of pixel-space operations and explore multi-tool coordination within our adaptive reasoning paradigm.
>
> ---
> **W4: Technical contribution seems weak.**
>
> We appreciate the reviewer's concern and would like to clarify the novelty and technical depth of our method. We introduce **the first framework that enables adaptive pixel-space reasoning, allowing VLMs to determine when pixel-level operations are necessary rather than applying them indiscriminately**. Our training framework **does not rely on any external pixel-level supervision or hand-crafted rules**, allowing the model to estimate the necessity of pixel-level operations directly from its own reasoning process. **Reviewer G9zR, RUxc and Xr3E** praise that our method is **intuitive and experiment results show the effectiveness of the method**. The reward design that considers both **correctness and alignment** with estimated necessity shows **thoughtful engineering and with novel method design.**
>
> **(1) Our contribution goes beyond “reward + rollout modifications”**
>
> The core technical innovations are:
>
> (a) First general framework for adaptive pixel-space reasoning
>
> Existing pixel-space reasoning always uses tools; **none learns when tools are truly necessary.** We introduce:
>
> - necessity estimation rollouts
> - tool-necessity alignment reward
> - rollout consistency reward
>
> **This formulation is new, principled, and solves a previously unaddressed problem.**
>
> (b) Model learns pixel-necessity from its own reasoning
>
> Our framework **removes the need for human labeling of which queries require zoom-in, predefined rules and auxiliary detectors or planners**. This self-estimation mechanism (comparing controlled rollouts) is a novel learning paradigm for tool-use LLMs.
>
> (c) Emergent adaptive behavior
>
> Our ablations demonstrate:
>
> - SFT alone → no adaptivity
> - RGRL → learns difficulty-aware tool usage
> - Highest accuracy + lowest tool ratio simultaneously
>
> **This emergent property comes directly from our algorithmic design.**
>
> **(2) Strong empirical advantages**
>
> Our method achieves SOTA on all five benchmarks, reduces tool usage by 66.5% on HR-Bench 4K and improves accuracy despite fewer operations. This indicates the framework is **both practically powerful and technically impactful**. We will strengthen the technical discussion and clarify why each component is necessary and novel.

---

### Note · Authors · 2025-12-22

I have read and agree with the venue's withdrawal policy on behalf of myself and my co-authors.